# A Two-Stage Service Migration Algorithm in Parked Vehicle Edge Computing for Internet of Things

**DOI:** 10.3390/s20102786

**Published:** 2020-05-14

**Authors:** Shuxin Ge, Meng Cheng, Xin He, Xiaobo Zhou

**Affiliations:** 1Tianjin Key Laboratory of Advanced Networking (TANK), College of Intelligence and Computing, Tianjin University, Tianjin 300350, China; cecilge@tju.edu.cn (S.G.); xiaobo.zhou@tju.edu.cn (X.Z.); 2School of Information Science, Japan Advanced Institute of Science and Technology (JAIST), Nomi, Ishikawa 923-1292, Japan; m-cheng@jaist.ac.jp; 3School of Computerand Information, Anhui Normal University, Wuhu 241001, China

**Keywords:** internet of things, parking duration, service provider, Lyapuunov optimization, Hungarian algorithm

## Abstract

Parked vehicle edge computing (PVEC) utilizes both idle resources in parked vehicles (PVs) and roadside units (RSUs) as service providers (SPs) to improve the performance of vehicular internet of things (IoT). However, it is difficult to make optimal service migration decisions in PVEC networks due to the uncertain parking duration and resources heterogeneity of PVs. In this paper, we formulate the service migration of all the vehicles as an optimization problem with the objective of minimizing the average latency. We propose a two-stage service migration algorithm for PVEC networks, which divides the original problem into the service migration between SPs and the serving PV selection in parking lots. The service migration between SPs is transformed to an online problem based on Lyapunov optimization, where the expected parking duration of PVs is utilized. A modified Hungarian algorithm is proposed to select the PVs for migration. A series of simulation experiments based on the real-world vehicle traces are conducted to verify the superior performance of the proposed two-stage service migration (SEA) algorithm as compared with the state-of- art solutions.

## 1. Introduction

The development of the automotive industry and the popularity of private cars have led to various issues such as traffic congestion, accidents, air pollution, etc. [1,2]. To ensure the safety and quality of experience (QoE), many novel vehicular Internet of Things (IoT) services are designed to control vehicle behavior by collecting the data from on-board sensors, for example, surrounding vehicle perception and collision risk [3]. In order to cope with great demands for these services, roadside units (RSUs) that are generally located in different geographic regions can be deployed on an edge server to act as a service provider (SP) [4]. However, the RSUs are always overloaded with the rapidly increasing number of vehicles and various service requests [5]. As such, more available resources are on demand [6]. It has been observed that the idle resources in parked vehicles (PVs) can be explored to efficiently provide services to other vehicles [7]. Therefore, a new computing paradigm, called parked vehicle edge computing (PVEC), has emerged to improve the vehicular IoT performance [8].

Generally speaking, private cars usually spend most of the time in parking lots with their resources in idle states, where the average parking duration is approximately seven hours [9]. PVs may consume energy for providing services to other vehicles, a discount in parking fees can be an incentive to a PV [10]. PVEC can take the parking lots as SPs by managing and integrating the rich resources such as storage resources and computing resources of the PVs [11]. The access point of the parking lot determines which PV is suitable to provide services to other vehicles. Through the PVEC, a vehicle can request for service from the parking lot to improve QoE. As such, the load of RSUs will be reduced, which means RSUs can provide higher QoE for vehicles [12]. However, it is a complicated process to manage the vehicle services with involvement of the parking [13,14]. For example, when the PV providing the service to other vehicles leaves the parking lot or refuses to continue providing the service, the ongoing service will be interrupted and delayed [15]. It is the case that the PVs with longer parking duration will help the stability of QoE. Moreover, different available resources in PVs, e.g., energy, also effect the network performance. When the resources can not support the service requests, or the PVs enter and leave the parking lots, the service needs to be migrated into the optimal PV so that the service continuity can be maintained [16]. With the migration, the service resumes exactly where it stopped at the target serving node. Furthermore, the service migration is also triggered by the vehicle mobility, which requires that the service is migrated to the nearest SP, for instance, the next immediate vehicle [17].

A proper service migration strategy should make a trade-off between the service latency and energy consumption of PVs, which encounters two challenges [18,19]. One challenge is that the parking duration of PVs greatly affects the overall performance of the service migration strategy [4]. As the leaving time of a PV cannot be known in advance, the service which is migrated to the leaving PV will result in an invalid migration or frequent migrations. Another challenge is that different long-term energy constraints in PVs are coupled with the service migration decision over the time, where the decisions have to be made without future information [20]. If a myopic migration decision achieves an optimal performance in terms of latency with huge energy consumption of a PV, the PV would suffer from energy shortage in the near future [21].

To address these challenges, this paper investigates the dynamic service migration in PVEC. A novel two-stage service migration (SEA) algorithm is proposed to solve the formulated problem. The main contributions of this paper are summarized as follows:We investigate the service migration problem in PVEC, and formulate the service migration process as a mixed integer nonlinear programming (MINP) to minimize the average service latency, where the QoE constraints and the available energy of PVs are taken into account. To our best knowledge, this is the first work that studies service migration in PVFC networks.We propose an efficient SEA algorithm to solve the formulated problem, where the original problem is divided into service migration between SPs and the selection of PVs for migration in parking lots. Utilizing Lyapunov optimization, we perform service migration decision between SPs in an online fashion without requiring future information. Meanwhile, a modified Hungarian algorithm is proposed to effectively tackle the difficult in selection of PVs with different available energy.The effectiveness of the proposed SEA algorithm is verified by simulation based on the real-world vehicle traces. Compared with three benchmarks, the simulation results demonstrate SEA algorithm has the superior performance in terms of average latency, utilization of PVs and energy consumption, especially with large total number of vehicles.

The rest of this paper is organized as follows. Section 2 reviews the relevant existing literature. The system model considered in the paper is presented in Section 3. Section 4 proposes SEA algorithm and conducts the complexity analysis. Section 5 discusses the performance of SEA algorithm. Section 6 concludes the paper.

## 2. Related Works

PVEC is an extension of edge network to relieve the high load pressure of RSUs in IoT. According to [22,23], the total number of PVs tend to remain stable with an average parking duration of around 6 h in North America. Many existing works design a incentive mechanism to encourage PVs to share its idle resources [18]. Malandrino et al. [24] leverages optimization model, where PVs serve as relay nodes, to maximize both freshness of the content that download retrieve and the efficiency in the utilization of radio resources. Balen et al. [25] utilize PVs that are near to passing vehicles to distribute messages. The objective of these methods is minimizing the download delay by choosing optimal PVs. Tonguz et al. [26] design a local rule to cache services in PVs, which are selected by biologically inspired self-organizing network approach, and thus achieve low latency of download updates. Li et al. [4] classify PVs into different types according to their parking duration. They design a contract-based incentive mechanism to motivate PVs with long parking duration to contribute their idle resources, and thus maximize the utility of the SP. However, these methods did not fully utilize the resources of PVs.

Recently, the majority of the literature on PVEC have focused on allocating the resources of PVs [7,27]. Gu et al. [13] propose a two-tier data center architecture that leverages the excessive storage resources in parking lots. They design three management policies for data center, i.e., non-replication, simple replication and network coding based replication, which aim to minimize the total communication cost in a closed form. Zhang et al. [28] study task assignment problem in PVs based on Stackelberg game. Through the game between PVs and tasks, it decide the serving PV for each task, and thus minimize the overall cost of a task publisher and subjective dissatisfaction caused by un-offloaded workloads. Huang et al. [29] minimize the service latency by introducing a machine learning framework, which allocates the heterogeneous resources of PVs based on the requirements of the services. Ouyang et al. [30] introduce a scheme based on Lyapunov optimization to minimize the system costs over time in compliance with the time-varying resources of SPs. Zhu et al. [31] make a trade-off between the average service latency and the overall quality loss by designing an event-triggered dynamic allocation scheme based on particle swarm optimization. However, these methods did not take the mobility of vehicles into account. As the vehicle moves, the SP of the vehicle changes constantly, therefore the latency performance obtained by these resource allocation schemes can not be maintained.

To tackle the mobility of vehicles, service migration has been extensively studied in edge networks to reduce the service latency [32,33]. Ksentini et al. [34] assume the user makes the rectilinear translation, and build a one-dimension (1D) MDP model to predict the trajectory. According to the predicted trajectory, the users determine the target serving node of the service migration. Nadembeg et al. [35] select optimal serving node by predicting the throughput in the target node. The service request is assumed to be split into several parts, and the service can be served by many servers at the same time. To be closer to the real scenes, Wang et al. [33] build a 2D Markov Decision Process (MDP) model to predict the user movement behavior in cellular networks, and further make the service migration to achieve the lowest system cost. Yu et al. [36] study a single-user service migration algorithm to select a fraction of the services according to the priority queue, and only migrates these services to reduce the negative effect of trajectory prediction error. Although these service migration strategies have a good performance in edge networks, it can not be directly applied to PVEC networks due to the uncertain parking duration and resources heterogeneity of PVs.

## 3. System Model

### 3.1. System Overview

In this paper, we consider a PVEC networks consisting *V* vehicles and *N* SPs, as shown in Figure 1. As vehicle A moves along the main road, it connects to SP 1, SP 2 and SP 3 sequentially. Initially, a service is deployed on SP 1 which can be accessed by vehicle A directly. As vehicle A approaches SP 2 and SP 3, a proper service migration decision will be made to decide whether to migrate the service by considering both the migration cost and response delay. As shown in the Figure 1, the service is migrated to SP 2 when vehicle A is near SP 2, and is not migrated to SP 3 when vehicle A is near SP 3. In the latter case, vehicle A gets access to the service by request routing. We divide the total time into *T* time slots, where each time slot lasts for π seconds. In time slot t∈[0,T−1], the vehicles move among the coverage regions. In general, a vehicle connects to the SP with the largest indicator of received signal strength. As the vehicle moves, each vehicle v∈[1,V] generates its service request randomly to SP n∈[1,N]. In each time slot, the operator decides whether and where to migrate the services for the vehicles.

By deploying the vehicular services, SPs can provide services to the vehicles within their radio ranges. When the corresponding service is deployed by the SP which the vehicle is connected with, the service request can be served locally with and minimized latency. These SPs are connected to each other via Ethernet so that any vehicle is able to access all the services along the request routing once it is in the coverage region of a SP.

Assume the SPs contain *I* RSUs and *J* parking lots (I + J = N), where SPs i∈[1,I] are RSUs with an edge server and other SPs j∈[I+1,I+J] are the parking lots equipped with a access point which is used to communicate with the PVs. The parking lots use the idle resources (e.g., storage capacity) of PVs to serve the vehicles. Let $K_{t}^{j}$ denote the set of PVs in parking lot *j*, which is updated by
(1)$$K_{t+1}^{j}=K_{t}^{j}\cup K_{t,in}^{j}-K_{t,out}^{j},$$
where $K_{t,in}^{j}$ and $K_{t,out}^{j}$ are the set of the PVs which enter the parking lot *j* in time slot *t*, and the set of the PVs which leave the parking lot *j* in time slot *t*, respectively. Thus, we have
(2)$$K_{t,in}^{j}\cap K_{t}^{j}=Ø,\;K_{t,out}^{j}\subseteq K_{t}^{j}.$$

Each PV $k\in K_{t}^{j}$ provides its idle resources to get a discount for its parking fees.

There are *M* different vehicular services that the vehicles may request for. We use a 5-tuple $<\lambda_{m},\gamma_{m},D_{m},f_{m,v},\theta_{m,v}>$ to describe service *m*, where the five coordinates denote the request data size, the computation intensity (i.e., CPU cycles/bit), the response deadline, computing capacity and storage requirement of service *m* for vehicle *v*, respectively. What a vehicle requests for multiple services is the same as that multiple vehicles request for a service. We assume each vehicle *v* continuously requests for one service m(v)∈[1,M] in this paper. Table 1 summarizes the key parameter notations in our paper.

### 3.2. Service Placement

The vehicle states are composed of its connected SP information, its serving node information in each time slot *t*, and the migration decision made by the networks operator.

For each vehicle *v*, we use $c_{v}\left(t\right)$ to denote the number of its connected SP. Thus, we have
(3)$$c_{t}\left(v\right)\in \left\{1,2,\cdots ,N\right\},\forall t,v.$$

Let $b_{t}\left(v\right)$ denote the serving node of vehicle *v* for service m(v) in time slot *t*, where $b_{t}\left(v\right)=n$ means vehicle *v* requests for service from SP *n* in time slot *t*. In addition, we use $b_{t}^{\prime }\left(v\right)$ to represent the migration decision, where the $b_{t}^{\prime }\left(v\right)=n$ means the service m(v) of vehicle *v* migrates to SP *n* in time slot *t*. Thus, the SP in the next time slot is updated by
(4)$$b_{t+1}\left(v\right)=b_{t}^{\prime }\left(v\right).$$

Similar to (Equation 3), we have
(5)$$\left\{ \begin{array}{ll} \; & b_{t}\left(v\right)\in \left\{1,\cdots ,N\right\},\\ \; & b_{t}^{\prime }\left(v\right)\in \left\{1,\cdots ,N\right\}, \end{array}\forall t,v.\right.$$

Moreover, the service request to SP *j* will be further allocated to the PVs. We define a $p_{t}\left(v\right)$, where $p_{t}\left(v\right)=k$ represents PV *k* in parking lot *j* which provides the service to vehicle *v* in time slot *t*. Meanwhile, we use $p_{t}^{\prime }\left(v\right)$ to represent the migration decision, where $p_{t}^{\prime }\left(v\right)=k$ represents the service of vehicle *v* is migrated to PV *k* in parking lot *j* at time slot *t*. The PV for migration in the next time slot is updated by
(6)$$p_{t+1}\left(v\right)=p_{t}^{\prime }\left(v\right).$$

Thus, we have
(7)$$\left\{ \begin{array}{lll} \; & p_{t}^{\prime }\left(v\right)=0, & b_{v}^{\prime }\left(t\right)\leq I,\\ \; & p_{t}^{\prime }\left(v\right)\in K_{t}^{j}, & \mathrm{otherwise}. \end{array}\right.$$

### 3.3. Service Latency

The service latency is composed of transmission latency, computing latency and migration latency. The transmission latency refers to the time it takes to send a request to the service node. The mobile edge network state in time slot *t* is described by an *N*-by-*N* matrix denoted by $R_{t}$. The (a,b)th (a∈[1,N],b∈[1,N]) element $R_{t}\left(a,b\right)$ denotes the available channel resources between SP *a* and SP *b* in time slot *t*. When a=b, element $R_{t}\left(a,b\right)$ is the positive infinity. Otherwise, it is randomly distributed in $\left(\eta_{min},\eta_{max}\right)$.

Meanwhile, we use ${\bar{R}}_{t}\left(a,b\right),\left(a\in \left(I,N\right],b\in K_{t}^{j}\right)$ to denote the transmission rate between the access point in parking lot *a* and PV *b* in time slot *t*, which is randomly distributed in $\left(\eta_{min},{\bar{\eta }}_{max}\right)$. As the distance from the access point and the PV is short, ${\bar{\eta }}_{max}$ is much higher than $\eta_{max}$.

Moreover, since the transmission latency between the vehicle and SP is hard to calculate precisely due to the movement of the vehicles, we use a constant *C* to approximate it. Assuming vehicle *v* generates a request to service m(v) at the beginning of time slot *t*, the request transmission latency includes the transmission latency between vehicle *v* and SP $b_{t}\left(v\right)$ in time slot *t*
(8)$$l_{1}^{T}\left(v,t\right)=\frac{\lambda_{m(v)}}{R_{t}\left(c_{t}\left(v\right),b_{t}\left(v\right)\right)}+C,$$
and the transmission latency between SP $b_{t}\left(v\right)$ and PV $p_{t}\left(v\right)$ in time slot *t*
(9)$$l_{2}^{T}\left(v,t\right)=\{\begin{array}{lll} \; & \frac{\lambda_{m(v)}}{{\bar{R}}_{t}\left(b_{t}\left(v\right),p_{t}\left(v\right)\right)}, & b_{t}\left(v\right)>I.\\ \; & 0, & \mathrm{otherwise}. \end{array}$$

The total transmission latency is
(10)$$L_{T}\left(v,t\right)=l_{1}^{T}\left(v,t\right)+l_{2}^{T}\left(v,t\right).$$

The computation latency of vehicle *v* for service m(v) is
(11)$$L_{C}\left(v,t\right)=\frac{\lambda_{m(v)}\gamma_{m(v)}}{f_{m(v),v}}.$$

Note that service m(v) may be migrated from $p_{t}\left(v\right)$ in SP $b_{t}\left(v\right)$ to $p_{t}^{\prime }\left(v\right)$ in SP $b_{t}^{\prime }\left(v\right)$ in time slot *t*. The migration latency includes the migration latency between SP $b_{t}\left(v\right)$ and SP $b_{t}^{\prime }\left(v\right)$
(12)$$l_{1}^{M}\left(v,t\right)=\frac{\theta_{m(v),v}}{R_{t}\left(b_{t}\left(v\right),b_{t}^{\prime }\left(v\right)\right)},$$
the migration latency between SP $b_{t}\left(v\right)$ and PV $p_{t}\left(v\right)$
(13)$$l_{2}^{M}\left(v,t\right)=\{\begin{array}{lll} \; & \frac{\theta_{m(v),v}}{{\bar{R}}_{t}\left(b_{t}\left(v\right),p_{t}\left(v\right)\right)}, & b_{t}\left(v\right)>I\\ \; & 0, & \mathrm{otherwise} \end{array}$$
and the migration latency between SP $b_{t}^{\prime }\left(v\right)$ and PV $p_{t}^{\prime }\left(v\right)$
(14)$$l_{3}^{M}\left(v,t\right)=\{\begin{array}{lll} \; & \frac{\theta_{m(v),v}}{{\bar{R}}_{t}\left(b_{t}^{\prime }\left(v\right),p_{t}^{\prime }\left(v\right)\right)}, & b_{t}^{\prime }\left(v\right)>I\\ \; & 0, & \mathrm{otherwise} \end{array}$$

The total migration latency is
(15)$$L_{M}\left(v,t\right)=\{\begin{array}{l} 0,\;\;b_{t}\left(v\right)=b_{t}^{\prime }\left(v\right),p_{t}\left(v\right)=p_{t}^{\prime }\left(v\right)\\ l_{1}^{M}\left(v,t\right)+l_{2}^{M}\left(v,t\right)+l_{3}^{M}\left(v,t\right),\mathrm{otherwise}. \end{array}$$

Thus, the service latency for vehicle *v* in time slot *t* is
(16)$$L\left(v,t\right)=L_{T}\left(v,t\right)+L_{C}\left(v,t\right)+L_{M}\left(v,t\right).$$

### 3.4. Energy Consumption

When entering the parking lot, PV *k* messages its available energy $\hat{e}\left(k,j\right)$ to the access point in parking lot *j*. The energy consumption in PV must satisfy this available energy constraint. The transmission energy consumption of PV *k* for vehicle *v* in parking lot *j* at time slot *t* is
(17)$$e_{T}\left(v,k,j,t\right)=P\pi \left(b_{t}\left(v\right),j\right)\pi \left(p_{t}\left(v\right),k\right)l_{2}^{T}\left(v,t\right),\;k\in K_{t}^{j},$$
where
(18)$$\pi \left(x,y\right)=\{\begin{array}{lll} \; & 1, & x=y\\ \; & 0, & \mathrm{otherwise}, \end{array}$$
where *P* is the communication power. The computing energy consumption of PV *k* for vehicle *v* in parking lot at time slot *t* is
(19)$$e_{C}\left(v,k,j,t\right)=\pi \left(b_{t}\left(v\right),j\right)\pi \left(p_{t}\left(v\right),k\right)\kappa f_{m(v),v}^{3}L_{C}\left(v,t\right)\;k\in K_{t}^{j},$$
where κ is the unit energy consumption per CPU cycle. The migration energy consumption of PV *k* for vehicle *v* in parking lot *j* at time slot *t* is
(20)$$\begin{array}{c} e_{M}\left(v,k,j,t\right)=P[\pi \left(b_{t}\left(v\right),j\right)\pi \left(p_{t}\left(v\right),k\right)l_{2}^{M}\left(v,t\right)+\pi \left(b_{t}^{\prime }\left(v\right),j\right)\pi \left(p_{t}^{\prime }\left(v\right),k\right)l_{3}^{M}\left(v,t\right)],\;k\in K_{t}^{j}. \end{array}$$

The total energy consumption in PV *k* for vehicle *v* in parking lot *j* at time slot *t* is
(21)$$e\left(v,k,j,t\right)=e_{T}\left(v,k,j,t\right)+e_{C}\left(v,k,j,t\right)+e_{M}\left(v,k,j,t\right),\;k\in K_{t}^{j}.$$

Thus, the available energy of PV *k* in parking lot *j* at the beginning of time slot *t* is
(22)$$E\left(k,j,t\right)=\{\begin{array}{lll} \; & \hat{e}\left(k,j\right), & t=\tau \left(k\right),\;k\in K_{t}^{j}\\ \; & E\left(k,j,t-1\right)-\sum_{v=1}^{V}e\left(v,k,j,t-1\right), & t\in \left(\tau \left(k\right),\;\bar{\tau }\left(k\right)\right],\;k\in K_{t}^{j} \end{array},$$
where τ(k) and $\bar{\tau }\left(k\right)$ are the enter time of PV *k* and the corresponding leaving time, respectively.

## 4. Two-Stage Service Migration Algorithm

### 4.1. Problem Formulation

Our goal is to minimize the long-term average latency with the energy consumption constraint in PVs by selecting the optimal migration strategy $b_{v}^{\prime }\left(t\right)$ and $p_{v}^{\prime }\left(t\right)$, which can be expressed as
$$\begin{array}{cc} P1:\; & \min_{b_{v}^{\prime },p_{v}^{\prime }}\lim_{T\to \infty }\frac{1}{T}\sum_{t=0}^{T-1}\sum_{v=1}^{V}L\left(v,t\right) \end{array}$$
(23)$$\begin{array}{cc} s.t.\; & L\left(v,t\right)\leq D_{m(v)},\;\forall v,t, \end{array}$$
(24)$$\begin{array}{l} \sum_{v=1}^{V}e\left(v,k,j,t\right)\leq E\left(k,j,t\right),\;\forall k,j,t,\\ \;(3),(5),(7). \end{array}$$

Constraint (Equation 23) ensures the service request can be completed before its response deadline. Constraint (Equation 23) denotes the energy of PV *k* in parking lot *j* can support the service request. The lack of future information is the difficulty in deriving the optimal solution to the above problem as solving P1 optimally requires the complete information. Moreover, P1 is an MINP problem that is proven to be NP-hard.

To solve P1, first we decouple P1 into two sub-problems, service migration between SPs problem and serving PV selection problem. Then we propose a two-stage algorithm, SEA, to solve these two sub-problems. In the first stage, we solve the service migration between SPs problem, i.e., deciding the serving SP for each vehicle. We utilize the Lyapunov optimization to convert this problem to per-slot optimization problem, which can be solved in an online style requiring only the current slot information. In the second stage, we solve the serving PV selection problem, i.e., deciding the serving PVs for vehicles whose serving SP is parking lot. We propose a modified Hungarian algorithm to choose the suitable PVs for vehicles, which fully utilizes the resources of all the PVs.

### 4.2. Problem Decomposition

The difference between RSUs and parking lots is that the parking lot needs to allocate its received service request from the vehicles to PVs with the energy constraint. Thus, we first consider parking lot *j* as a special RSU, which produces a extra latency during the request process. Replacing the transmission rate $\bar{R}\left(b_{t}\left(v\right),p_{t}\left(v\right)\right)$ in (Equation 9) and (Equation 13), and the transmission rate $\bar{R}\left(b_{t}^{\prime }\left(v\right),p_{t}^{\prime }\left(v\right)\right)$ in (Equation 14) with a constant $\frac{{\bar{\eta }}_{max}}{2}$, we can obtain a new total latency ${\bar{L}}_{1}\left(v,t\right)$. Meanwhile, the available energy in parking lot *j* at the beginning of time slot *t* is
(25)$$\begin{array}{r} \bar{E}\left(j,t\right)=\sum_{k\in K_{t}^{j}}E\left(k,j,t\right). \end{array}$$

Note that this energy only can be used before the PVs leave the parking lots. The first stage can be described by
$$\begin{array}{rr} P2:\; & \min_{b_{v}^{\prime }}\sum_{t=0}^{T-1}\sum_{v=1}^{V}{\bar{L}}_{1}\left(v,t\right) \end{array}$$
(26)$$\begin{array}{ll} s.t.\; & {\bar{L}}_{1}\left(v,t\right)\leq D_{m(v)},\forall v, \end{array}$$
(27)$$\begin{array}{l} \sum_{k\in K_{t}^{j}}\sum_{v=1}^{V}e\left(v,k,j,t\right)\leq \bar{E}\left(j,t\right),\\ \;(3),(5). \end{array}$$

After obtaining the solution of P2, solving the original problem P1 only needs to select the serving PV for the received migrated service in each parking lot *j* at time slot *t*. The total service latency in parking lot *j* is
(28)$${\bar{L}}_{2}^{j}\left(v,t\right)=\pi \left(b_{t}\left(v\right),j\right)[l_{2}^{T}\left(v,t\right)+l_{C}\left(v,t\right)+l_{2}^{M}\left(v,t\right)]+\pi \left(b_{t}^{\prime }\left(v\right),j\right)l_{3}^{M}\left(v,t\right).$$

Let $V_{t}^{j}$ denote the set of the vehicles, which migrate the service to parking lot *j* at time slot *t*, i.e., $b_{t}^{\prime }\left(v\right)=j$. In this way, the second stage problem for parking lot *j* in time slot *t* can be represented by
$$\begin{array}{ll} P3:\; & \min_{p_{v}^{\prime },\;v\in V_{t}^{j}}\sum_{v\in V_{t}^{j}}{\bar{L}}_{2}^{j}\left(v,t\right)\\ s.t.\;\; & \left(5),(24\right). \end{array}$$

The detail of the SEA algorithm is shown in Algorithm 1. In each time slot *t*, we obtain the migration decision $b_{v}^{\prime }$, i.e., the SP of each vehicle by solving the first stage problem (Line 2). Based on this, we obtain the migration decision $p_{v}^{\prime }\left(t\right)$, i.e., actual serving PV of the vehicle which migrates the service to the PV in parking lot *j* by solving the second stage problem (Lines 3–5).

**Algorithm 1** Two-stage service migration (SEA) algorithm.**Input :**$c_{v}\left(t\right),b_{v},b_{v}^{\prime }$;
**Output :** Service migration decision $b_{v}^{\prime },p_{v}^{\prime }$;
 1: **for**
t=0 to T−1
**do**
 2:  Obtain $b_{v}^{\prime }\left(t\right)$ by solving P2 according to Algorithm 2; 3:  **for**
j=I+1 to *N*
**do** 4:   Obtain $p_{v}^{\prime }\left(t\right),\;v\in V_{t}^{j}$ by solving P3 according to Algorithm 3; 5:  **end for** 6: **end for**

### 4.3. Service Migration between SPs

In this subsection, we transform the P2 to per-slot optimization problems based on Lyapunov optimization.

#### 4.3.1. Expected Parking Duration

The energy consumption under a certain migration decision greatly influences the available energy in the future, e.g., the parking lot needs to store the energy to attend frequent requests. We predict the average parking duration to ensure the energy consumption in each time slot is reasonable, in order to avoid rapidly running out of energy and lack of energy in the future, and also avoid using too little energy and compromising, the current latency performance.

Firstly, we identify the type of the PVs based on the parking duration. The parking duration is sorted in an ascending order and classified into $\bar{N}$ types $\left\{\delta_{1},\delta_{2},\cdots ,\delta_{\bar{N}}\right\}$, given by
(29)$$\delta_{1}<\cdots <\delta_{\bar{n}}<\cdots <\delta_{\bar{N}},\;\bar{n}\in \left\{1,2,\cdot ,\bar{N}\right\}.$$

However, the information asymmetry between the SP and the PVs causes the SP can not realize the type of PVs. We generate the probability distribution of PV types through the historical data statistics. The probability that a PV belongs to type-$\bar{n}$ in parking lot *j* is denoted by $\mathrm{Pr}(\delta_{\bar{n}})$, and
(30)$$\sum_{\bar{n}=1}^{\bar{N}}\mathrm{Pr}\left(j,\delta_{\bar{n}}\right)=1.$$

The expected parking duration of PVs in parking lot *j* is
(31)$$E\left\{\tau \left(j\right)\right\}=\sum_{\bar{n}=1}^{\bar{N}}\mathrm{Pr}\left(j,\delta_{\bar{n}}\right)\delta_{\bar{n}}.$$

#### 4.3.2. Algorithm for the Service Migration between SPs

As discussed in Section 4.1, to solve P2 online, we need to convert P2 to per-slot optimization problems requiring only current slot information. Thus, according to the expected parking duration of PVs, we leverage the Lyapunov optimization technique and build a virtual queue to guide the migration decisions with the energy constraint.
(32)$$Q_{j}\left(t+1\right)={\left[Q_{j}\left(t\right)+\sum_{k\in K_{t}^{j}}\sum_{v=1}^{V}e\left(v,k,j,t\right)-\frac{\bar{E}\left(j,t\right)}{E\{\tau (j)\}}\right]}^{+},$$
where $Q_{j}\left(t\right)$ is the queue backlog for parking lot *j* in time slot *t*, indicating the deviation of the current energy consumption from the energy constraint. The Lyapunov function is defined as L(ξ(t)), representing the “congestion level” in energy deficit queue, which is
(33)$$L\left(\xi \left(t\right)\right)≜\frac{1}{2}\sum_{j=I+1}^{N}{\bar{Q}}_{j}^{2}\left(t\right).$$

A small value of L(ξ(t)) implies that the queue backlog is small, i.e., the virtual queue has strong stability. To keep the energy deficit queue stable, i.e., to enforce the energy consumption constraints by persistently pushing the Lyapunov function towards a lower value, let Δ(t) denote the one-slot Lyapunov drift, which is Δ(t)=L(ξ(t+1))−L(ξ(t)), and we have
$$\begin{array}{l} \Delta \left(t\right)=\frac{1}{2}\sum_{j=I+1}^{N}[Q_{j}^{2}\left(t+1\right)-Q_{j}^{2}\left(t\right)]\\ \leq \frac{1}{2}\sum_{j=I+1}^{N}{\left[\sum_{k\in K_{t}^{j}}\sum_{v=1}^{V}e\left(v,k,j,t\right)-\frac{\bar{E}\left(j,t\right)}{E\{\tau (j)\}}\right]}^{2}+\sum_{j=I+1}^{N}Q_{j}\left(t\right)[\sum_{k\in K_{t}^{j}}\sum_{v=1}^{V}e\left(v,k,j,t\right)-\frac{\bar{E}\left(j,t\right)}{E\{\tau (j)\}}]\\ \leq JU+\sum_{j=I+1}^{N}Q_{j}\left(t\right)[\sum_{k\in K_{t}^{j}}\sum_{v=1}^{V}e\left(v,k,j,t\right)-\frac{\bar{E}\left(j,t\right)}{E\{\tau (j)\}}], \end{array}$$
where
(34)$$U≜\sup\frac{1}{2}\sum_{j=I+1}^{N}{\left[\sum_{k\in K_{t}^{j}}\sum_{v=1}^{V}e\left(v,k,j,t\right)-\frac{\bar{E}\left(j,t\right)}{E\{\tau (j)\}}\right]}^{2}.$$

Under the Lyapunov optimization framework, the underlying objective of our optimal control decision is to minimize a supremum bound on the following drift-plus-cost expression in each time slot.
$$\Delta \left(t\right)+\mu \sum_{v=1}^{V}\bar{L}\left(v,t\right)\leq JU+\sum_{j=I+1}^{N}Q_{j}\left(t\right)\sum_{k\in K_{t}^{j}}\sum_{v=1}^{V}e\left(v,k,j,t\right)+\mu \sum_{v=1}^{V}\bar{L}\left(v,t\right),$$
where the parameter μ>0 can make the trade-off between the service latency and the energy consumption of SPs.

Under the Lyapunov optimization framework, the underlying migration decision of P2 is made by solving the optimization problem P4
$$\begin{array}{ll} P4:\; & \min_{b_{v}^{\prime }}\mu \sum_{v=1}^{V}{\bar{L}}_{1}\left(v,t\right)-\sum_{j=I+1}^{N}Q_{j}\left(t\right)\sum_{k\in K_{t}^{j}}e\left(k,j,t\right)\\ s.t. & \;\;\;(26),(3),(5). \end{array}$$

Note that P4 only requires the current available information. By considering the additional term $\sum_{j=I+1}^{N}Q_{j}\left(t\right)\sum_{k\in K_{t}^{j}}\sum_{v=1}^{V}e\left(v,k,j,t\right)$, the energy deficit of SPs during the current slot is taken into account. A high value of $\bar{E}\left(j,t\right)$ means it is critical to minimize the energy deficit. The detail of solving the service migration between SPs is shown in Algorithm 2. We first initialize the PV set as a null set (Line 1). At the beginning of the time slot *t*, the parking lots update its PVs set (Line 3). After this step, we can obtain $b_{v}^{\prime }$ by solving P4 in an iterative manner (Line 4). Next, we calculate the available energy in the next time slot, and update the energy queue (Lines 5, 6). Finally, the solution in the time slot *t* of P4 becomes the initial state in the next time slot t+1 (Line 7).

**Algorithm 2** Service migration between the service providers (SPs) algorithm.**Input :**$Q_{j}\left(t\right)\leftarrow 0,\bar{E}\left(j,0\right)\leftarrow 0,c_{v}\left(t\right),b_{v},b_{v}^{\prime }$;**Output :** Service migration decision $b_{v}^{\prime }$; 1: initialize the PV set $K_{0}^{j}=Ø$; 2: **for**
t=0 to T−1
**do** 3: Update the PV set $K_{t}^{j}$ according to (Equation 1); 4: Obtain $b_{t}^{\prime }$ by solving P4; 5: Update $Q_{j}\left(t+1\right)$ according to (Equation 32); 6: Update $\bar{E}\left(j,t+1\right)$ according to (Equation 25), 7: Update $b_{v}\left(t+1\right)$ according to (Equation 5); 8: **end for**


According to [21], we find that the average latency satisfies
(35)$$\lim_{T\to \infty }\frac{1}{T}\sum_{t=0}^{T-1}\sum_{v=1}^{V}{\bar{L}}_{1}\left(v,t\right)\leq \frac{JU}{\mu }+O^{*}.$$
where $O^{*}$ is the optimal value of P2. Meanwhile, the energy consumption in parking lot *j* satisfies
(36)$$\sum_{k\in K_{t}^{j}}\sum_{v=1}^{V}e\left(v,k,j,t\right)\leq \frac{\bar{E}\left(j,t\right)}{E\{\tau (j)\}}+\frac{U}{\epsilon^{*}}+\frac{\mu }{J\epsilon^{*}}\left(O^{max}-O^{*}\right),$$
where $O^{max}$ represents the long-term latency achieved by no migration strategy. It demonstrates an [O(1/μ),O(μ)] latency-energy trade-off. The latency performance of P4 improves with μ→∞. However, it also results in the large energy deficit queue. That is, the energy consumption in PVs grows linearly when μ increases.

### 4.4. Serving PV Selection

After solving P2, we can know the vehicles which request service from parking lots. In each time slot *t*, we need to select the serving PVs for these vehicles according to the service latency and available energy in PVs. A possible solution is to randomly choose a PV for each vehicle. However, the latency performance may not be optimal. Thus, we propose a modified Hungarian algorithm, which can fully utilize the resources of PVs in each parking lot, to solve P3.

First of all, we build a cost function F(v,k,j,t) to calculate the fitness of a migration decision $p_{t}^{\prime }\left(v\right)$ (i.e., $k=p_{t}^{\prime }\left(v\right)$). A low value of F(v,k,j,t) means the vehicle gains a big benefit when it is served by PV *k*.
(37)$$F\left(v,k,j,t\right)=\alpha {\bar{L}}_{2}^{j}\left(v,t\right)-[\frac{E(k,j,t)}{\Delta_{k}}-e\left(v,k,j,t\right)],$$
where $\Delta_{k}$ is the number of service requests that PV *k* offers. α>0 is the weight parameter. The modified Hungarian algorithm is shown in Algorithm 3. It works as follows.
Compare the the number of service requests of vehicles $S(V_{t}^{j})$ with the number of available PVs $S(K_{t}^{j})$, and let $Z=\max\{S\left(V_{t}^{j}\right),\;S\left(K_{t}^{j}\right)\}$ (Line 1).$Z>S(K_{t}^{j})$ means the PVs are not enough to achieve one-to-one service. We generate a expanded set $K_{t}^{j*}$, which size is equal to *Z*. When $Z=S(K_{t}^{j})$, $K_{t}^{j*}=K_{t}^{j}$ (Lines 2–10).If $Z>S(V_{t}^{j})$, we generate a expanded vehicle set $V_{t}^{j*}$, $V_{t}^{j*}=V_{t}^{j}$ otherwise (Lines 11–14).Next, we initialize a $S(V_{t}^{j})$-by-$S(K_{t}^{j})$ cost matrix F, where each element $F\left(v,k\right),v\in V_{t}^{j*},k\in K_{t}^{j*}$ is the cost value F(v,k,j,t) (Line 15). In this way, we can obtain the migration decision $p_{v}^{\prime }\left(t\right)$ by traditional Hungarian algorithm (Line 16).

**Algorithm 3** Modified Hungarian algorithm.**Input :**$V_{t}^{j},K_{t}^{j}$;**Output :** Service migration decision $p_{v}^{\prime }$; 1: Initialize $Z=\max\{S\left(V_{t}^{j}\right),\;S\left(K_{t}^{j}\right)\}$; 2: Let $K_{t}^{j*}=K_{t}^{j}$; 3: **while**
$Z>\sum_{k\in K_{t}^{j}}\Delta_{k^{*}}$
**do** 4: Find the maximum $\frac{E(k^{*},j,t)}{\Delta_{k^{*}}}$; 5: $\Delta_{k^{*}}=\Delta_{k^{*}}+1$; 6: **end while** 7: **for**
$k\in K_{t}^{j}$
**do** 8: Make $\Delta_{k}$ copies of the the PV *k*; 9: Add the copies of PV *k* into $K_{t}^{j*}$; 10: **end for** 11: Let $S\left(V_{t}^{j*}\right)=S\left(V_{t}^{j}\right)$ 12: **while**
$Z>S(V_{t}^{j})$
**do** 13: Add $Z-S(V_{t}^{j})$ virtual vehicles $v^{*}$ into $S(V_{t}^{j*})$, which the cost function $F(v^{*},k,j,t)=0$; 14: **end while** 15: Initialize cost matrix F according to (Equation 37) based on $V_{t}^{j*},K_{t}^{j*}$; 16: Obtain $p_{t}^{\prime }\left(v\right)$ according to traditional Hungarian algorithm;


The time-complexity of the modified Hungarian algorithm relies on the difference between cost function of PVs for the vehicles. When the cost function values are same, the time-complexity becomes higher, which is at most $O(S^{3}\left(V_{t}^{j*}\right))$.

## 5. Performance Evaluation

In order to evaluate the performance of SEA algorithm, we conduct simulation experiments. We build a PVEC system with 10 RSUs and 5 parking lots, deployed in the map. PVs behavior is formulated based on a real dataset which are collected by SmartParking App [37]. Figure 2 shows the cumulative distribution function of the parking duration for PVs. The vehicle trajectories are collected from the real-world Rome taxi traces obtained in 2014 [38]. Four typical vehicular services, i.e., emergency stop, collision risk, accident report and parking, are used as the service that the vehicles may request for. The parameters of the services are listed in Table 2. The other major parameters are listed in Table 3.

We compare SEA with three benchmarks.

Never-migrate scheme (NM): The service is deployed in the original SP and never migrates to other SPs. Note that all the service in PVs randomly migrate to other PVs when the PV leaves the parking lot.Always Migration (AM): the service is always migrated to the nearest SP, while the service in the parking lot is assigned to the PVs that can provide best latency performance regardless of the energy consumption constraint in PVs.Dynamic Markov Decision Process (DMDP) [33]: This is a single-user service migration algorithm, where the serving RSU or PV is predicted using the MDP model. The optimal service migration decision is made to minimize the energy consumption. DMDP were applied to each of the users independently in our simulations.

### 5.1. Impact of Different Total Number of Vehicles

Figure 3 shows the average latency with different total number of vehicles. It was observed that the average latency with NM is the highest, which is owing to a significant amount of latency for request transmission. On the other hand, the lowest average latency is achieved by AM, as AM always migrate service to the nearest SPs and further migrate service to PV that can provide best latency performance. Since the objective of DMDP is to minimize the energy cost, the average latency of DMDP is higher than that of AM and of SEA. Meanwhile, with the total number of vehicles increases from 100 to 1000, the average latency greatly increases to reduce the migration energy consumption. The average latency of SEA is quite close to that of AM when the total number of vehicles is smaller than 500. When the total number of vehicles exceeds 500, SEA slowly grows with the limited energy of PVs. However, it is still very close to AM.

Figure 4 shows the average utilization rate of PVs with different total number of vehicles. As can be seen from the figure, the utilization rate of PVs for NM is lowest. This is expected as the services were only migrated when the serving PV was leaving the parking lot. Due to the service requests concentrating on PVs that can provide best latency performance, the utilization rate of PVs of AM is near-minimal. Meanwhile, the utilization rate of PVs of DMDP maintain a second highest performance. It was observed that the highest average utilization rate is achieved by SEA as the modified Hungarian algorithm takes the advantage of available energy of PVs. The average utilization rate of PVs with SEA improves by up to 22.2%, as compared with the state-of-the-art solution.

Figure 5 shows the average additional energy consumption, which is the sum of the energy consumption exceeding the available energy in PVs divided by total number of PVs, with different total number of vehicles. The highest average additional energy consumption is achieved by AM, where increases exponentially with the total number of vehicles increases from 100 to 1000. This is because too many service requests are concentrated on a PV, resulting in the rapid depletion of energy in this PV. Meanwhile, the average additional energy consumption of NM is near to highest, which is caused by a significant amount of energy being required for requesting transmissions. The average additional energy of DMDP is near to optimal with the MDP model can not predict the parking duration of PV. The average additional energy consumption of SEA is near to zero when the vehicle number below 600. Even with the total number of vehicles increases from 600 to 1000, the average additional energy consumption grow slowly.

Figure 6 shows the average surplus energy in PVs, which is the sum of the residual energy in PVs divided by total number of PVs, with different total number of vehicles. Note that if the energy consumption in PV exceeds the available energy, the surplus energy is zero. The performance of NM, AM and DMDP are consistent with those in Figure 5. The highest utilization rate of PVs of SEA lead the energy in PVs is fully exploited. Thus, the average surplus energy of SEA reduce rapidly to 36.4 mJ with the total number of vehicles increase from 100 to 1000. Such observations above indicate that SEA algorithm is suitable for PVEC networks.

### 5.2. Impact of the Weight Parameter in the Cost Function

Figure 7 shows the average latency and average additional energy consumption with different α, while the total number of vehicle is 1000. By increasing α from 0.01 to 100, SEA algorithm puts more emphasis on the average latency, i.e., the average latency decreases and the average additional energy consumption increases. By adjusting the value of α, we can achieve a balance between these two. In our simulation experiments, we set α=10.

## 6. Conclusions

In this paper, we investigated a service migration problem in PVEC networks. We formulated the service migration process as an MINP optimization problem to minimize the average service latency while satisfying the energy constraints of PVs. A two-stage algorithm, called the SEA algorithm, was proposed to solve the MINP problem, which divides the original problem into the service migration between SPs and the selection of PVs for migration in parking lots sub-problems. Utilizing Lyapunov optimization, we perform service migration decision between SPs in an online fashion without requiring future information. Meanwhile, a modified Hungarian algorithm is proposed to effectively tackle the difficult in selection of PVs with different available energy and services. A series of simulation experiments based on the real-world traces were conducted to verify the superior performance in reducing the average latency with the energy constraint of PVs by our proposed SEA algorithm. It demonstrated that the proposed solution can reduce average latency by up to 17.2% and improve the average utilization rate of PVs by up to 22.2%, as compared with the state-of-the-art solution DMDP. It indicated that SEA algorithm is suitable for PVEC networks. Besides the available energy, storage and computing resources also affect the migration decision, which will be taken into account in future studies.

## Figures and Tables

**Figure 1 sensors-20-02786-f001:**
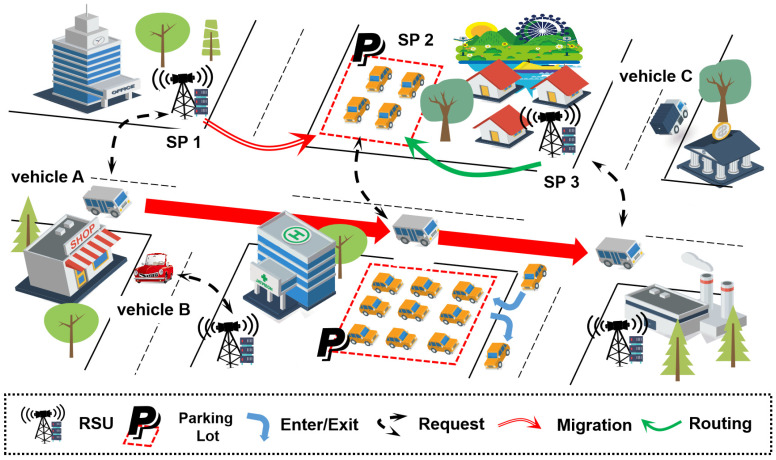
System overview.

**Figure 2 sensors-20-02786-f002:**
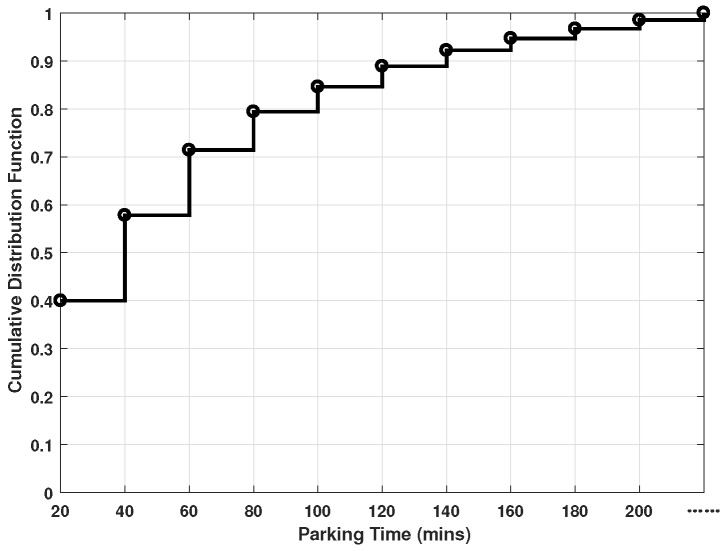
The cumulative distribution function of the parking duration for parked vehicles (PVs).

**Figure 3 sensors-20-02786-f003:**
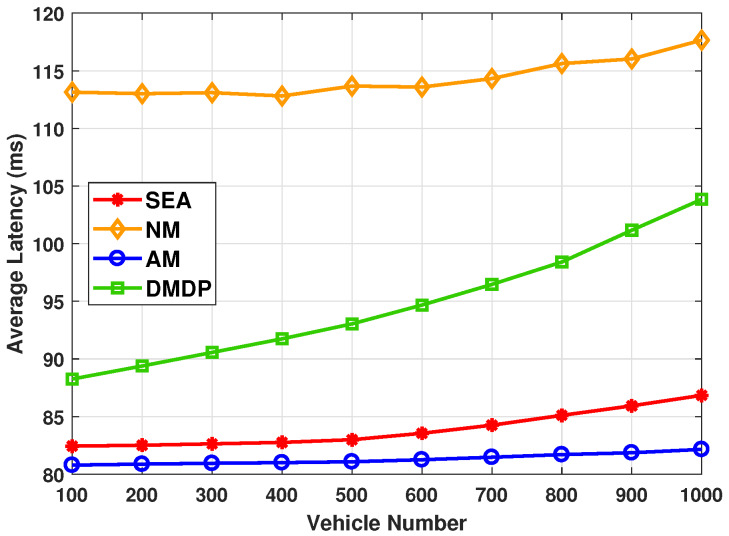
The average latency with different total number of vehicles.

**Figure 4 sensors-20-02786-f004:**
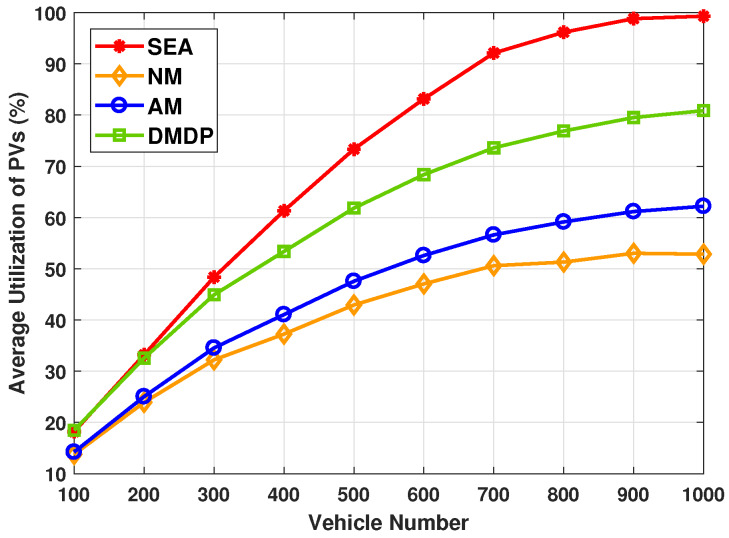
The average utilization rate of PVs with different total number of vehicles.

**Figure 5 sensors-20-02786-f005:**
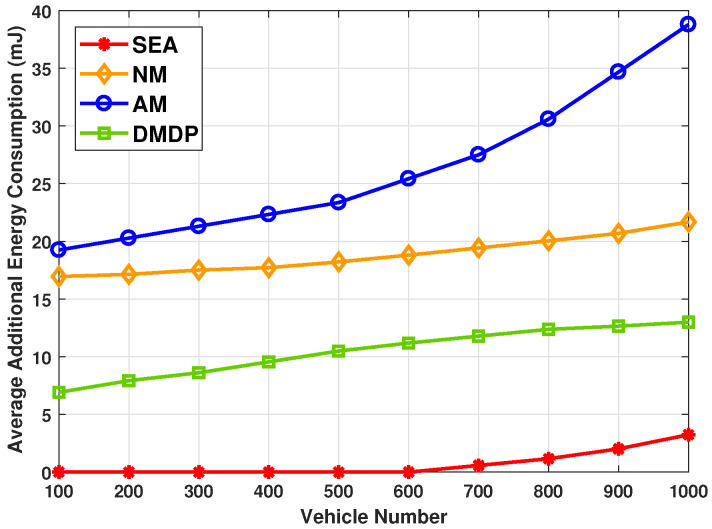
The average additional energy with different total number of vehicles.

**Figure 6 sensors-20-02786-f006:**
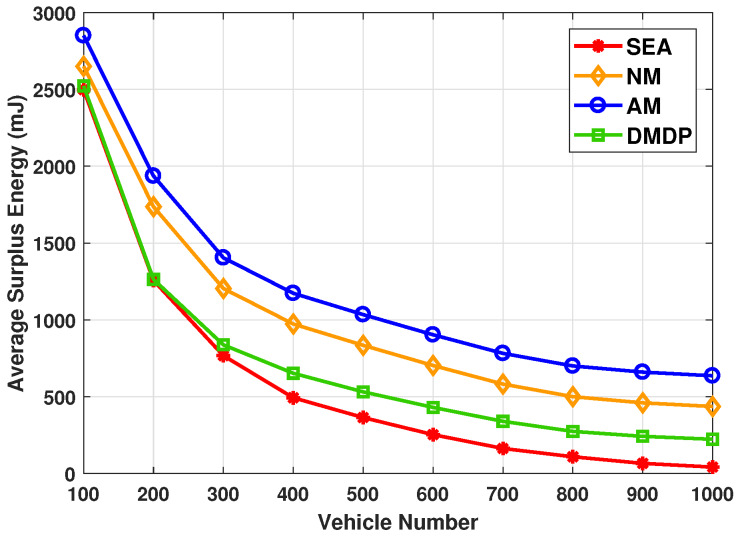
The average surplus energy with different total number of vehicles.

**Figure 7 sensors-20-02786-f007:**
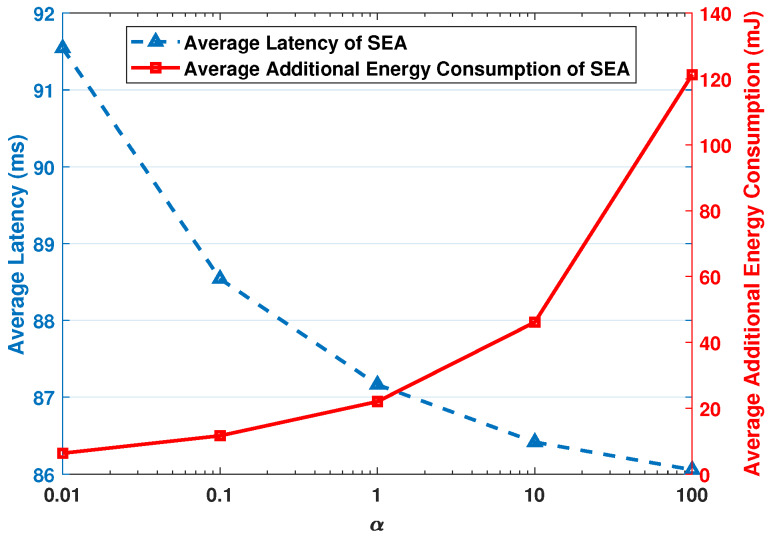
The average latency and average additional energy consumption with different α, while the total number of vehicles is 1000.

**Table 1 sensors-20-02786-t001:** Definitions of notations.

Notation	Definition
$K_{t+1}^{j}$	the set of PVs in parking lot *j*
$c_{t}v$	the connected SP of the vehicle *v* in time slot *t*
$b_{t}\left(v\right)$	the SP of the request of vehicles *v* in time slot *t*
$b_{t}^{\prime }\left(v\right)$	the migration decision of SP for vehicle *v* in time slot *t*
$p_{t}\left(v\right)$	the serving PV of the request of vehicles *v* in time slot *t*
$p_{t}^{\prime }\left(v\right)$	the migration decision of PV for vehicle *v* in time slot *t*
$\lambda_{m}$	the input data size of service *m*
$\gamma_{m}$	the computation intensity of service *m*
$D_{m}$	the response deadline of service *m*
$\Omega_{m}$	the priority of service *m*
$f_{m,v}$	the CPU cycle requirements of a service *m* request from vehicle *v*
$\theta_{m,v}$	the storage requirements of a service *m* request from vehicle *v*
$R_{t}\left(i,j\right)$	The transmit rate between SP *i* and SP *j* in time slot *t*
L(v,t)	the total latency of vehicle *v* in time slot *t*
*P*	the communication power
E(k,j,t)	the total energy consumption of PV *k* in parking lot *j* at time slot *t*
$\tau \left(k\right),\bar{\tau }\left(k\right)$	the enter time and leaving time of PV *k*

**Table 2 sensors-20-02786-t002:** Service parameters.

Type	$D_{m}$	$\lambda_{m}$	$\gamma_{m}$ (KB)	$f_{m,v}$ (GHz)	$\theta_{m,v}$ (MB)
emergency stop	0.1	3200	36	[1.6, 3.2]	[1, 3]
collision risk	0.1	4800	40	[2.4, 4.8]	[1, 3]
accident report	0.5	4800	28	[2.4, 4.8]	[2, 3]
parking	0.1	1200	80	[0.6, 1.2]	[3, 5]

**Table 3 sensors-20-02786-t003:** Experimental parameters.

Notation	Value
$\eta_{min}$	5 Mbps
$\eta_{max}$	10 Mbps
${\bar{\eta }}_{max}$	20 Mbps
κ	$10^{-26}$
*P*	0.5 W

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
