# Peer review of "A Two-Stage Service Migration Algorithm in Parked Vehicle Edge Computing for Internet of Things"

_sensors, 2020, doi:10.3390/s20102786_

Round 1

Reviewer 1 Report

A Two-Stage Service Migration Algorithm in Parked Vehicle Edge Computing for Internet of Things

Abstract

13: Define the acronym “SEA” before using it.

15: “Internet of thins, . . .” → “Internet of things, . . .”

Introduction

21-22: Separate the two example services mentioned with an and or an or in the statement “…,for example, surrounding vehicle perception, collision risk”

26-27: Reference works for the statement: “It has been observed that the idle resources in parked vehicles (PVs) can be explored to efficiently provide services to other vehicles”

66: “he effectiveness” → “The effectiveness”

69: “The simulation results demonstrate SEA algorithm has the superior performance.”, Based on what? Under what conditions?

17-73: The problem and justification of the work are correctly explained, but the methodology and the way in which the technologies used contribute to the construction of the solution remain to be clarified. Complement and expand the introduction a little more if possible. Related works are mentioned in a later chapter.

Related Works

75-77: What region of the world are these data from? Mention even if the reference is added.

74-115: Contrast the related works, advantages and disadvantages in relation to the proposal made in the article.

System Model

119: Figure 1 contains too many elements, it would help the reader to add a brief explanation of the elements and flows shown in the Figure, either in the paragraph or in the caption.

117-139: Could use a diagram to clarify the system overview and the elements that are involved, like what was tried with Figure 1, but more detailed.

141: Define the acronym “BS” before using it.

165-170: Chapter 3 presents a technical description of the model, reconsider the appearance of a problem formulation subsection in this chapter, or restructure it and move it to another chapter.

Two-Stage ServiceMigration Algorithm

176: Why did you decide to use the mentioned algorithm? Does it have advantages or disadvantages? Contrast with others?

Performance Evaluation

248-255: It could be used a table to present the parameters in a clearer way.

Conclusions

Most of the conclusions are a summary of what was done in the work. Specific conclusions should be emphasized based on the results obtained and the contribution made. Expand and complement the conclusions section.

In general, the chapters seem well structured and described.

Reviewer 2 Report

This paper studies the service migration for the service migration of computational tasks for edge data centers composed of parked vehicles in a parking lot.

The authors consider service latency (performance) and energy consumption (cost) as their main KPIs.

They propose an algorithm based on Lyapunov Optimization for service migration and Hungarian Method for service placement.

1) The authors need to perform a thorough English check. I found some typographical and grammatical errors.

Page1, line11... "the excepted parking time"

Page1, line13... "proposed SEA algorithm as": SEA abbreviation was not predefined

Page2, line66... "he effectiveness"

Page11, line244... "Hungarian algorithm relay on"

2) Why do you choose the Lyapunov method for Stage 1, service migration? What is your motivation?

3) Why do you choose the Hungarian method for Stage 2, service placement? What is your motivation?

4) What is the weakness of the proposed algorithm?

Reviewer 3 Report

Overall, I am quite happy with the most important aspect of this submission which concerns its methodology, problem focus, and scientific soundness.  I am also reasonably happy with the evaluation which is suitable to the problem at hand and sufficiently convincing.

The main problem or weaknesses of the submission concerns presentation.  There are a lot of linguistic problems which make the article difficult to read.  This applies to the entire presentation of content generally.  Here are some examples of issues:

  • (linguistic) "service migration ... are": grammatical number mismatch.
  • (clarity) "uncertain parking times of PVs": it this the time when the vehicle is parked or the DURATION it is parked for, or something else entirely?  I infer from the rest that it is the duration, in which case it is "uncertain duration of a vehicle being parked".
  • (typo) "internet of thins"

  • (style) keywords should not contain words from the title; these serve indexing purposes in ADDITION to the title.

  • (style) far (FAR!) too many abbreviations which make the paper difficult to read.

  • (linguistic) "collecting the data in on-board sensors": FROM on-board...

  • (linguistic) "There are two following challenges.": just bad English.

  • (clarity) "parking time of PVs": as before.

  • (typo) "he effectiveness of": THE

  • (clarity) "parking time is 6.64 hours on average [22].": as before

  • (style) "Obviously": avoid using this word; if something is obvious then why state it?

  • (linguistic): "number of vehicle": number of vehicles (this is very confusing because it leads the reader to think that this may refer to some label given to vehicles)

  • (clarity) I suggest you give an overview of the entire method before you delve into detail so that the reader knows ahead of time what problem you are addressing, how it fits the overall scheme, etc.

Round 2

Reviewer 3 Report

Criticism taken constructively, concerns addressed adequately.